# Oncogenic EFNA4 Amplification Promotes Lung Adenocarcinoma Lymph Node Metastasis

**DOI:** 10.3390/cancers14174226

**Published:** 2022-08-30

**Authors:** Xiangyu Zhao, Yuxing Chen, Xiaoqin Sun, Zaoke He, Tao Wu, Chenxu Wu, Jing Chen, Jinyu Wang, Kaixuan Diao, Xue-Song Liu

**Affiliations:** 1School of Life Science and Technology, ShanghaiTech University, Shanghai 201203, China; 2Shanghai Institute of Biochemistry and Cell Biology, Chinese Academy of Sciences, Shanghai 200031, China; 3University of Chinese Academy of Sciences, Beijing 100049, China

**Keywords:** EFNA4, lung cancer, lymph node metastasis, copy number alterations

## Abstract

**Simple Summary:**

Lymph nodes are likely to be the first stop for lung cancer metastasis. To further investigate the mechanism of lung cancer lymph node metastasis, we performed cancer genome analysis and found that EFNA4, a member of the ephrin (EPH) family, is amplified and up-regulated in lung tumor patients, especially in patients with lymph node metastases. In vitro and in vivo experiments show that overexpression of *EFNA4* promotes lung tumor cell proliferation and migration, whereas knockdown or knockout of *EFNA4* inhibits cell proliferation and migration. Altogether, our results suggest that the DNA amplification of the *EFNA4* genome locus could play an oncogenic function in promoting lung cancer lymph node metastasis.

**Abstract:**

Lymph nodes metastases are common in patients with lung cancer. Additionally, those patients are often at a higher risk for death from lung tumor than those with tumor-free lymph nodes. Somatic DNA alterations are key drivers of cancer, and copy number alterations (CNAs) are major types of DNA alteration that promote lung cancer progression. Here, we performed genome-wide DNA copy number analysis, and identified a novel lung-cancer-metastasis-related gene, *EFNA4*. The *EFNA4* genome locus was significantly amplified, and *EFNA4* mRNA expression was significantly up-regulated in lung cancer compared with normal lung tissue, and also in lung cancer with lymph node metastases compared with lung cancer without metastasis. *EFNA4* encodes Ephrin A4, which is the ligand for Eph receptors. The function of EFNA4 in human lung cancer remains largely unknown. Through cell line experiments we showed that *EFNA4* overexpression contributes to lung tumor cells growth, migration and adhesion. Conversely, *EFNA4* knockdown or knockout led to the growth suppression of cells and tumor xenografts in mice. Lung cancer patients with *EFNA4* overexpression have poor prognosis. Together, by elucidating a new layer of the role of EFNA4 in tumor proliferation and migration, our study demonstrates a better understanding of the function of the significantly amplified and overexpressed gene *EFNA4* in lung tumor metastasis, and suggests EFNA4 as a potential target in metastatic lung cancer therapy.

## 1. Introduction

Lung cancer remains the leading cancer-related mortality in men and women [1,2,3]. According to histological morphology, lung cancer can be divided into two main categories: non-small-cell carcinoma (NSCLC) and small-cell carcinoma. Additionally, NSCLC can be divided into lung adenocarcinoma (LUAD), lung squamous cell carcinoma (LUSC) and large cell lung cancer [4,5,6]. Clinical data have shown that LUAD incidence is higher than LUSC, and accounts for about 50% of all lung cancers [7]. Effective treatment strategies, including molecular targeted drugs and immune checkpoint inhibitors, can improve the outcome of patients with lung cancer [8]. However, limited therapeutic effect on patients with distant metastasis leads to poor prognosis. Metastasis of lung cancer can occur, even when the primary tumor is still small. Therefore, patients with advanced LUAD have a poor prognosis, with an average 5-year survival rate of less than 20% [9,10,11,12]. Further research on new therapeutic targets for LUAD is essential. The development of microarray and high-throughput sequencing technology has greatly promoted bioinformatics, which use of computers for processing the biological data, for example, form the Cancer Genome Atlas (TCGA). By mining these cancer genome data, new cancer-related genes can be effectively identified [13,14,15].

*EFNA4* is an important member of the ephrins family. Ephrins and ephrin receptors (Ephs) are membrane-binding proteins, forming a bidirectional and interactive signaling system [16,17,18]. Ephrins are membrane-bound proteins, divided into two families, class A and class B. The Ephrin A members are anchored to the membrane by glycosyl-phosphatidylinositol (GPI) whereas Ephrin B members possess a transmembrane segment. At the same time, their receptors (Ephs) were classified as EphA or EphB subfamily according to their preferential binding to Ephrin A or Ephrin B [19]. Previous studies have shown that members of the ephrin family play an important role in the pattern of nervous system, vascular system assembly and angiogenesis [20,21,22,23,24]. EFNA4 has also been reported to be involved in the proliferation of oral squamous cell carcinoma, hepatocellular carcinoma and gastric cancer [25,26,27]. However, the functions and mechanisms of *EFNA4* in lung cancer are still not clear.

We reported here the results of a systematic cancer genome analysis, and *EFNA4* locus amplification was identified as one of the somatic DNA alterations that exerted a potent lung cancer driving function. Moreover, we found that *EFNA4* overexpression is linked to lymph node metastasis in LUAD. Together, these results identified a novel oncogenic driving function of *EFNA4* in lung cancer metastasis, and implicate *EFNA4* as a potential target in lung cancer therapy.

## 2. Materials and Methods

### 2.1. Cell Culture and Cell Transfection

All cell lines were used for no more than 20 generations (or 4 months) after resuscitation, and mycoplasma evaluations were performed regularly. H1299, A549 and PC9 cells were cultured in RPMI 1640 (Gibco, Grand Island, NY, USA) supplemented with 10% FBS (Gemini, West Sacramento, CA, USA) and 1% penicillin–streptomycin (Gibco). HEK-293 cells were cultured in DMEM (Gibco) supplemented with 10% FBS (Gibco) and 1% penicillin–streptomycin (Gibco). All cells were cultured at 37 °C with 5% CO_2_. For plasmid transfection, 100 nM (final concentration) shRNA or 3~5 µg plasmid in 50 µL Opti-MEM (Gibco) was mixed with 5 µL transfection reagent (Lipofectamine, Invitrogen, Waltham, MA, USA) in 50 µL of Opti-MEM, then the mixture was incubated at room temperature for 30 min and subsequently added drop-wise to cells. After 6 h, the mixture was removed, and cells were incubated with fresh media.

### 2.2. DNA Constructs

The human *EFNA4* coding sequences was amplified and cloned into pcDNA3.1 vector (Invitrogen). pLKO.1-shEFNA4 was synthesized by GENEWIZ (Suzhou, China). Additionally, pLKO.1-shNC expressing luciferase was used as control. The plasmids sequence was verified by Sanger sequencing.

### 2.3. Lentivirus Production

Expression clone DNA (6 µg) was mixed with packaging plasmids pMD2.G (1.5 µg), psPAX2 (4.5 µg) in 6-well plates. EZ Trans transfection reagent (LIFE iLAB BIO, Shanghai, China) was mixed DNA mix and OPTI-MEM (Gibco) and incubated for 20~40 min at RT. Cells were allowed to settle for at least 2 h, before the DNA-EZ mix was added per well. Additionally, 10 h later, the medium was replaced with fresh medium. After 48 h, the virus was harvested and stored at −80 °C until use.

### 2.4. sgRNA Design

sgRNA sequences were designed using the Broad Institutes sgRNA designer tool (http://portals.broadinstitute.org/gpp/public/analysis-tools/sgrna-design (accessed on 9 September 2016)). The exon of *EFNA4* was selected for guiding RNA design and sgRNA was cloned into pGL3-U6. The binary vector constructed, containing two guides, gRNA1 and gRNA2, was used to transform competent *E. coli*. Positive clones were confirmed by Sanger sequencing.

### 2.5. RNA Extraction and Real-Time PCR

Total RNA was isolated using TRIzol reagent (Invitrogen). Total RNA (1 ug) was used for the synthesis of cDNA using M-MLV reverse transcriptase (Invitrogen). Real-time PCR was performed with SYBR Green (Bimake, Houston, TX, USA). All qPCR gene expression data were normalized against the housekeeping gene ACTB.

### 2.6. Immunoblotting

Cells were washed three times with cold PBS. RIPA lysate buffer mixed with protease inhibitor cocktail (100:1) was added. Proteins were separated by 12% SDS-PAGE. Additionally, separated proteins were transferred onto a polyvinylidene difluoride membranes. Anti-Ephrin A4 antibody (ab209058, Abcam, Cambridge, UK), Anti-β-actin antibody (AC-15, Sigma, St. Louis, MO, USA), were used for Western blot analysis. For chemiluminescence, horseradish peroxidase-conjugated secondary antibodies and Western Lightining^®^ Plus-ECL (NEL105001EA, PerkinElmer, Waltham, MA, USA) were used. Western blotting was performed as described previously [28].

### 2.7. Cell Proliferation Assay

Cells were seeded in 96-well plates at a density of 1000 cells per well, then left to grow for 7 days. MTT assay was used for evaluating cell viability as described previously [29].

### 2.8. Colony Formation Assay

The cells were seeded in 6-well plates, and each cell line was inoculated into 3 wells (1000 cells/well). Cells were incubated for 4 h at 37 °C with 5% CO_2_, 95% air and complete humidity. After incubation for 2 weeks, the culture medium was aspirated, and the cells were gently washed twice with 1× PBS. Cells were fixed by paraformaldehyde, and then stained with methyl blue solution. After extensive washing, the stained cell colonies were counted and photographed.

### 2.9. Migration Assay

Overall, 5 × 10^5^ cells/well were seeded in 6-well plates and each cell line was inoculated into 3 wells. Scratch wound assay creates a gap in confluent monolayer of tumor cells to mimic a wound. Then, images at 0 h were captured to record the initial area of the wounds, and the recovery of the wound healing was evaluated at 24, 48, 72, 96 h.

### 2.10. Cell Adhesion Assays

Overall, 1000 cells/well were seeded in 96-well plates, and each cell line was inoculated into 3 wells. The cells were cultured for 2/4/6 h and then washed with 1× PBS once. After 24 h, MTT assay was used for evaluating cell viability as described previously [29].

### 2.11. Confocal Laser Scanning Microscopy

The cells were plated overnight on 20 mm glass-bottom cell culture dishes. Cells were fixed with 4% formaldehyde for 20 min, and permeated by 0.5% Triton-x100 in PBS, then blocked with 1% bovine serum albumin in PBS. After washing with PBS, the cells were incubated with anti-EFNA4 antibody (E7061, Abclonal, Woburn, MA, USA) overnight at 4 °C in 1% bovine serum albumin in PBS. Alternatively, the negative control cells were incubated with 1% bovine serum albumin in PBS under the same conditions. After washing, the cells were re-stained with appropriate Alexa Fluor 488-conjugated secondary antibodies for 1 h at 37 °C. Cells were then stained with 0.1 μg/mL 4′,6-diamidino-2-phenylindole (DAPI) for 5 min at room temperature. The cells were analyzed by confocal microscopy using a Leica TCS SP8 laser scanning confocal microscope (Leica Microsystems, Wetzlar, Germany) with a 63× oil immersion objective. Images were acquired and processed using LAS AF Lite software.

### 2.12. Xenograft Model

All animal studies were approved by the Animal Care and Use Committee of National Center for Protein Science Shanghai. Male BALB/c nude mice, 4 weeks old, were purchased from Shanghai (China) SLAC Laboratory Animal Co. and maintained in microisolator cages. Tumor cells were injected subcutaneously (4 × 10^6^ cells in 100 µL PBS) into the flanks of mice (*n* = 3 per group). Tumor volumes were measured every 3 days, and volumes were calculated using the formula: V = L × W^2^/2, where V is the tumor volume, W is tumor width, and L is tumor length.

### 2.13. Lymphatic Metastasis and Survival Analysis of the High Versus Low Gene EFNA4 Expression Tumors

Clinical data, CNA data and mRNA expression data were downloaded from TCGA data portal (https://portal.gdc.cancer.gov/ (accessed on 8 October 2021)). Differential expression analysis was performed using UCSCXenaShiny [30] (https://hiplot-academic.com/advance/ucsc-xena-shiny (accessed on 10 October 2021)) and the results obtained are visualized by Hiplot [31] (https://hiplot.com.cn/basic (accessed on 12 October 2021)). The procedure for CNA analysis was adopted from our previously work [32] and GISTIC2.0 (https://anaconda.org/HCC/gistic2 (accessed on 8 October 2021)) was used to identify recurrent copy-number alterations. To compare high EFNA4 expression level versus low EFNA4-expression-level tumors, we considered the tumors having an EFNA4 expression level to be within the 30th percentile (low EFNA4 group) and the tumors having an EFNA4 expression level higher than the 70th percentile (high EFNA4 group). We used Fisher’s exact test to assess the difference in the lymphatic metastasis between the two groups. The Kaplan–Meier method and log-rank tests were used to assess overall survival (OS) and recurrence-free survival (RFS). A similar analysis was performed in GSE11969 and GSE41271 from public Gene Expression Omnibus (GEO) dataset.

### 2.14. HPA Analysis

The immunohistochemical (IHC) staining images of EFNA4 were obtained from the Human Protein Atlas (HPA) database (https://www.proteinatlas.org/ (accessed on 10 June 2022)) and the positive cells were quantified by ImageJ (https://imagej.nih.gov/ij/ (accessed on 10 June 2022)).

### 2.15. Statistical Analyses

For data analysis, the SPSS statistical package for Windows (SPSS 16), R program (4.0.5), the GraphPad Prism Software (version 5.01, Vienna, Austria) and Microsoft Excel (Microsoft Office 2013 for Windows, Albuquerque, NM, USA) were used. Results are presented as the mean ± standard error. Other statistical analysis was performed using the unpaired, two-tailed Student’s *t*-test and *p* < 0.05 was considered to indicate a statistically significant difference.

## 3. Results

### 3.1. EFNA4 mRNA Up-Regulation and CNV Amplification in Human Tumors

Lymph node metastasis is common in patients with NSCLC. In order to find new targets for effective treatment of metastatic lung cancer, we examined data from multiple cancer types from TCGA project and screened gene with high specific expression in metastatic tumors through cancer genome studies. First, GISTIC analyses [33] were performed in lung cancer to identify genomic regions that are significantly gained, and nine statistically significant “peak” regions of amplification in lung cancer were identified (Appendix A). Chromosome region 1q21.3 was found to be frequently amplified in lung cancer, especially in metastatic lung cancer. The analysis of the RNA-Seq data reveals six significantly up-regulated genes (fold change > 3, *p*-value < 0.001) in 1q21.3 between TCGA lung tumor samples and normal tissues (Appendix A). By Fisher’s exact analysis, overexpression of *EFNA4* had a strong correlation with lymph node metastasis in lung tumor (Figure 1A). As shown in Figure 1B, EFNA4 is highly expressed in multiple tumor types, especially in LUAD, compared with paired normal tissues. Then, we assessed *EFNA4* copy number variation (CNV) in patients and found that *EFNA4* is amplified in many tumor types (Figure 1C). Immunohistochemical data from the Human Protein Atlas (HPA) database also showed that EFNA4 was highly expressed in the tumor (Figure 1D). Thus, we speculated that *EFNA4* is a novel lung-cancer-metastasis-associated gene.

### 3.2. EFNA4 Overexpression Promotes the Proliferation and Migration of Lung Tumor Cells

To explore the relationship between high expression of *EFNA4* and occurrence and development of lung cancer, lentivirus-mediated *EFNA4*-overexpressed lung tumor cells (H1299, A549 and PC9) were constructed (Figure 2A). Interestingly, it was found that *EFNA4* overexpression enhanced the ability of colony formation (Figure 2B), cell proliferation (Figure 2C), migration (Figure 2D) and cell adhesion (Appendix A) in the lung cancer cell lines. To further investigate the function of *EFNA4* in vivo, we examined the tumorigenic ability of *EFNA4*-overexpression cells in immunodeficient mice. The stable *EFNA4*-overexpression A549 cell line and wild type cells were subcutaneously injected into the nude mice, and the corresponding transplanted tumor was formed after continuous feeding for 30 days, and the size and weight of each transplanted tumor were measured. In sharp contrast to control cells stable *EFNA4* overexpression A549 cells developed significantly bigger tumors (Figure 2E). RT-qPCR analyses of these tumor xenografts validated the over-expression of *EFNA4* (Appendix A). Together, these results indicate that *EFNA4* can promote the proliferation and migration of lung tumor cells.

### 3.3. EFNA4 Depletion Inhibits the Growth and Migration of Lung Tumor Cells

In order to further verify our conclusion, we generate *EFNA4* knockout cells using CRISPR/Cas9 system. *EFNA4* was knocked out with two sgRNAs in A549 and PC9 cells (Appendix A), which was confirmed by Western blotting and PCR (Figure 3A and Appendix A). In the Cancer Dependency Map project (DepMap, www.depmap.org (accessed on 5 October 2021), knockout of *EFNA4* led to decreased growth in most cancer cell lines (Figure 3B). Consistent with DepMap results, our results show that *EFNA4* deletion significantly affected cell viability (Figure 3C), and also suppressed cell migration, invasion and adhesion Figure 3D,E and Appendix A). To access the in vivo effect of *EFNA4* knockout on lung cancer cells, *EFNA4* knockout A549 cells were xeno-grafted in nude mice via subcutaneous injection. Additionally, *EFNA4* knockout significantly decreased the total tumor weights and size (Figure 3F). Two shRNA targeting *EFNA4* were designed, and stable EFNA4 shRNA cell lines of A549 and PC9 were constructed. The expression level of *EFNA4* in the stable cell line were tested by qPCR, and it was found that shRNA strongly inhibits *EFNA4* expression (Appendix A). As shown in Appendix A, shRNA-mediated down-regulation of *EFNA4* resulted in decreased number of colonies. At the same time, it was found that *ENFA4* knockdown by shRNA inhibits cancer cell proliferation (Appendix A), and attenuates cancer cell migration (Appendix A). Similar results were obtained with two different cell lines, A549 and PC9. Taken together, these results suggest that deletion of *EFNA4* can inhibit the growth and migration of lung tumor cells.

### 3.4. Ephrin A4 Was Located on the Cell Membrane

Previous studies reported that ephrins are anchored to the cell membrane by glycosylphosphatidylinositol (GPI). EFNA4 is a member of the ephrins family. To investigate the location of Ephrin A4, immunofluorescence with anti-ephrin A4 antibody was performed. As shown in Appendix A, overexpression of *EFNA4* leads to a significant increase in fluorescence intensity around the cell membrane. EFNA4−depleted cells show weak fluorescence signal around the membrane (Appendix A). Most likely due to the limited specificity of anti-ephrin A4 antibodies, some non-specific fluorescence appeared in the cytoplasm. Through the above experiments, we concluded that Ephrin A4 was located on the cell membrane.

### 3.5. EFNA4 Could Be a Potential Prognosis Marker in LUAD

Based on TCGA, the expression level of *EFNA4* in LUAD and its relationship with prognosis, overall survival (OS) and recurrence-free survival (RFS), were analyzed. *EFNA4* was found to be negatively associated with OS and RFS of LUAD patients (Figure 4A). This result was also validated by two independent datasets (Figure 4B).

## 4. Discussion

Lung cancer has the highest incidence and mortality among major cancers throughout the world, and LUAD patients account for a large proportion of lung cancer patients. Despite the current multiple therapies, the prognosis and treatment for LUAD patients, particularly for those with lymph node metastasis, remain poor, and new therapeutic targets need to be identified. Therefore, the purpose of the current study was to identify and validate genes that are involved in the progression of metastatic LUAD.

Based on the above starting points, we have three main findings. First, this study provides the first bioinformatics and also experimentally validated link between *EFNA4* and lymph node metastases in lung cancer. Second, our experimental evidence suggests that *EFNA4* may be an important player in lymph node metastasis, and targeting this protein represents a method to treat lung cancer. An anti-ephrin-A4 antibody–drug conjugate, PF-06647263, achieved sustained tumor regression in triple-negative breast and ovarian tumor-initiating cells [34]. However, in a phase I study, patients treated with PF-06647263 showed limited response [35]. This result demonstrates the complexity of *EFNA4* in tumor therapy. On the one hand, targeting EFNA4 does show anti-tumor activity; on the other hand, patients show limited responses and numerous side effects. Therefore, we still need to explore the application scene and molecular mechanisms of *EFNA4*. Third, EFNA4 could be a prognostic marker which can be used to evaluate a lung cancer patient’s clinical outcome.

A limitation of this study is the lack of molecular mechanisms by which EFNA4 promotes lung cancer metastasis. The present evidence indicates that signaling by EPHA2 and its ligand, EFNA4, promotes hepatocellular carcinoma cell migration and the combination of EFNA4 and EPHA10 promotes proliferation and migration in oral squamous cell carcinoma cells [25,26]. In fact, during the study, we also found the positive relationships between EFNA4 and key glycolytic enzymes (not shown in this paper). In addition to glycolysis, EFNA4-high tumor cells are more metabolically active than EFNA4-low tumor cells in many other metabolic pathways, including the tricarboxylic acid cycle, oxidative phosphorylation, fatty acid metabolism, and so on. Further works are needed to explain the relationship between EFNA4 and metabolic reprogramming in tumors. Based on cancer genomic analysis, EFNA4 overexpression is associated with lung cancer lymph node metastasis. Direct experimental evidence demonstrating the function of EFNA4 in lung cancer lymph node metastasis is still lacking.

Overall, here, we show that the up-regulation of *EFNA4* could promote the cell growth, migration and adhesion of lung tumor cells, whereas the down-regulation of *EFNA4* led to the opposite results. Additionally, tumor survival analysis revealed significant impact of *EFNA4* expression on survival, and *EFNA4* overexpression is linked to poorer lung cancer outcome. Our work suggests that EFNA4 could be targeted by therapeutic agents and be a prognostic marker for lung cancer.

## 5. Conclusions

This study provides new insights into the link between EFNA4 and lung cancer metastasis. The amplified and up-regulated gene *EFNA4* is closely associated with lymph node metastasis. Additionally, we experimentally show that EFNA4 promotes the growth and migration of lung cancer cells, and knockout of *EFNA4* leads to the inhibition of lung tumor cell proliferation and migration. Based on the prognosis and survival analysis, EFNA4 would be a novel biomarker to predict the progression of lung cancer patients. Together, our work suggests that EFNA4 may be a potential therapeutic target and prognostic marker in lung cancer.

## Figures and Tables

**Figure 1 cancers-14-04226-f001:**
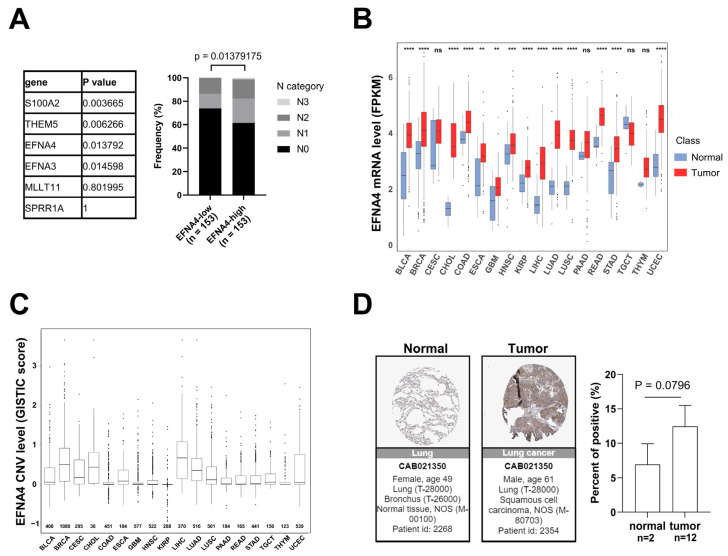
*EFNA4* expression and copy number variation (CNV) status in human cancers. (**A**) The relationship between the expression level of 6 upregulated genes at 1q21.3 and the stage of lymph node metastasis. Gene expression data and related clinical information of patients were obtained from TCGA project. *p* values were determined using Fisher’s exact test; (**B**) *EFNA4* mRNA level in various types of human tumors compared with normal tissues. Gene expression data were obtained from TCGA project. The box shows median value and 25th and 75th percentiles; (**C**) *EFNA4* copy number values (GISTIC scores) obtained from GISTIC2 software in cancers are shown based on TCGA database. GISTIC2 CNV value 0 means normal copy number. The box shows median value and 25th and 75th percentiles; (**D**) IHC images of the EFNA4 in LUAD and normal tissues obtained from the HPA database. The positive cells were quantified by ImageJ. *p*-values are represented as ns (not significant), ** *p* < 0.01, *** *p* < 0.001 and **** *p* < 0.0001.

**Figure 2 cancers-14-04226-f002:**
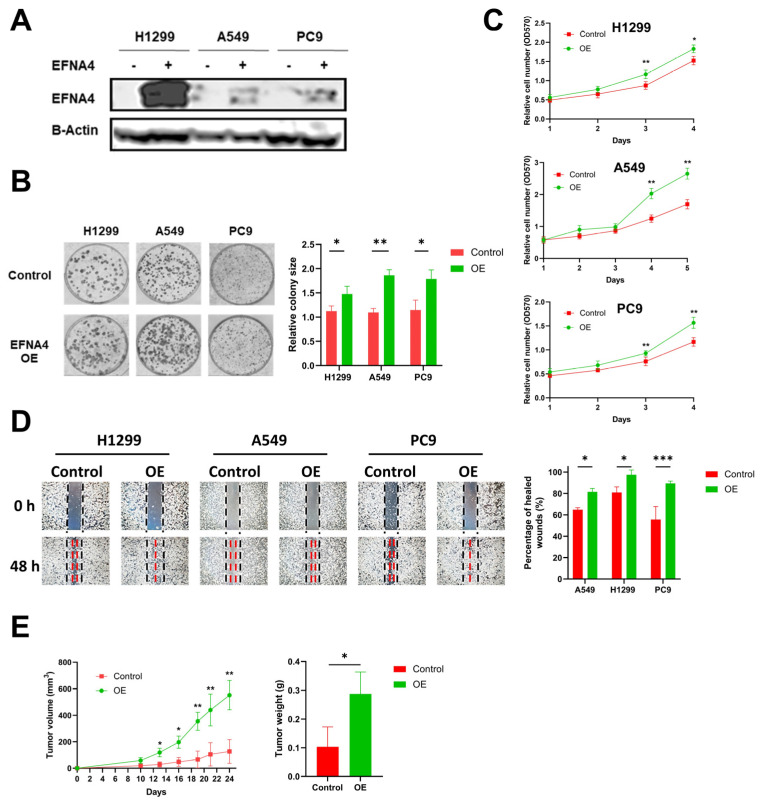
*EFNA4* overexpression can promote the proliferation and migration of lung tumor cells. (**A**) Western blot analysis for EFNA4 in H1299, A549 and PC9 control cells and *EFNA4*-overexpression (OE) cells that were transfected with *EFNA4* expression virus vector; (**B**) representative images of colony formation of H1299, A549 and PC9 *EFNA4*-overexpressing cells and control cells. Colonies were stained with methylene blue; (**C**) MTT cell proliferation assay. H1299, A549 and PC9 control cells and *EFNA4*-overexpressing cells proliferation was assayed by MTT method. Cell survival rates were analyzed at 1, 2, 3, 4 and 5 days. Data are expressed as mean ± SD of three independent experiments; (**D**) representative images of scratch wound healing assay of control and *EFNA4*-overexpressing lung tumor cells; (**E**) volume and wet weight of xenograft tumors. *EFNA4*-overexpressing and control A549 cells were subcutaneously implanted in nude mice (*n* = 3). Data are expressed as mean ± SD of three independent experiments. OE: overexpression. Significant *p*-values are represented as * *p* < 0.05, ** *p* < 0.01 and *** *p* < 0.001.

**Figure 3 cancers-14-04226-f003:**
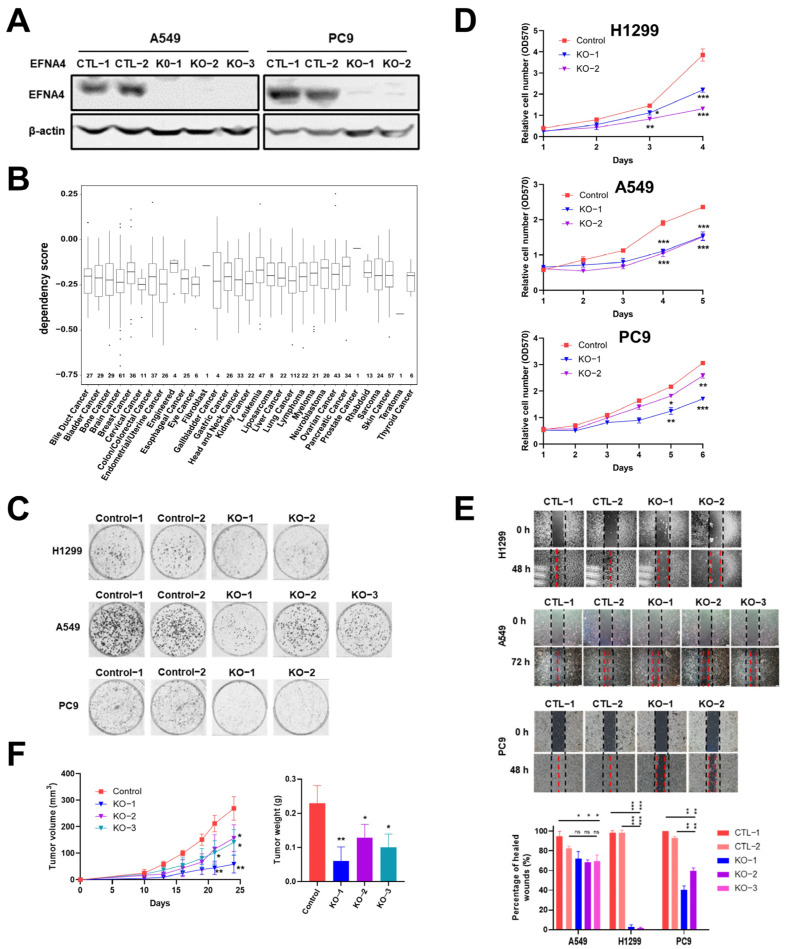
The deletion of *EFNA4* can inhibit the growth and migration of lung tumor cells. (**A**) Western blot confirmation of *EFNA4* knockout in A549, PC9 cells; (**B**) box plot of the DepMap (CRISPR) gene dependency score for *EFNA4*. Numbers above the X-axis represent respective sample sizes. Lower and upper parts of the box correspond to the 25th and 75th percentiles, respectively; (**C**) colony formation assay. Representative image of control and *EFNA4*-KO established colonies (*n* = 3); (**D**) MTT cell proliferation assay. A549 and PC9 control cells and *EFNA4* knockout cells proliferation was measured by MTT method; (**E**) cell migration analysis by wound-healing assay in *EFNA4* knockout cells and control cells of A549 and PC9 cells; (**F**) volume and wet weight of xenograft tumors. *EFNA4* knockout and control A549 cells were subcutaneously implanted in nude mice (*n* = 3). Data represented as mean ± SD of three independent experiments. CTL: control; KO: knockout. *p*-values are represented as ns (not significant), * *p* < 0.05, ** *p* < 0.01 and *** *p* < 0.001.

**Figure 4 cancers-14-04226-f004:**
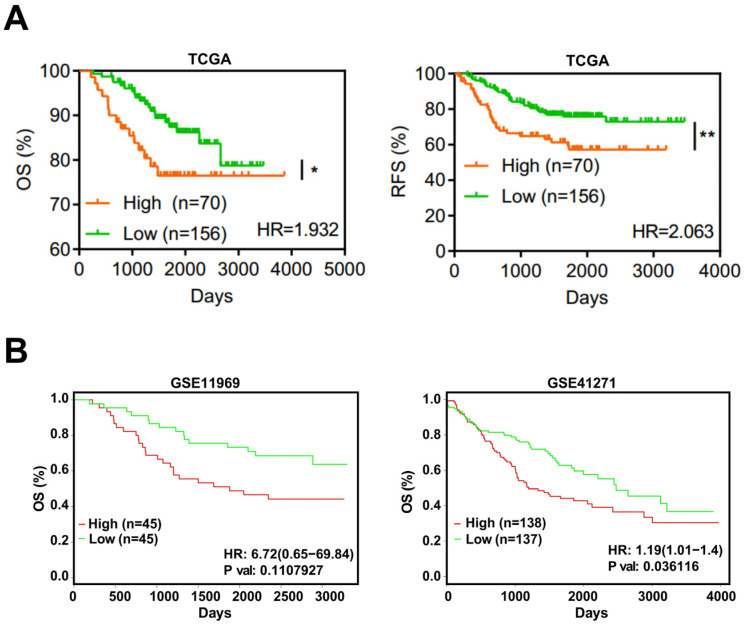
*EFNA4* expression is correlated with the survival rate in lung cancer patients. (**A**) Kaplan–Meier curves of overall survival (OS, left) rate and the recurrence survival (RFS, right) rate in TCGA LUAD patients with high *EFNA4* expression (*n* = 70) and low *EFNA4* expression (*n* = 156). (**B**) Kaplan–Meier overall survival curves of patients with lung tumor in two independent NCBI GEO cohorts (GES11969, GSE41271). Patients are separated into two groups based on *EFNA4* mRNA level. Significant *p*-values are represented as * *p* < 0.05 and ** *p* < 0.01.

## Data Availability

Clinical data, CNA data and mRNA expression data were downloaded from TCGA data portal (https://portal.gdc.cancer.gov/ (accessed on 8 October 2021). IHC images were downloaded from HPA database (www.proteinatlas.org (accessed on 10 June 2022). A subset of the clinical data included in the manuscript is publicly available at NCBI GEO: https://www.ncbi.nlm.nih.gov/geo/query/acc.cgi?acc=GSE41271 (accessed on 10 June 2022) and https://www.ncbi.nlm.nih.gov/geo/query/acc.cgi?acc=GSE11969 (accessed on 10 June 2022).

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
