# Peer review of "Oncogenic EFNA4 Amplification Promotes Lung Adenocarcinoma Lymph Node Metastasis"

_cancers, 2022, doi:10.3390/cancers14174226_

Round 1

Reviewer 1 Report

Overall nice study that used several tools to show the effect of the overexpression of EFNA4 on the proliferation, migration of lung tumor cells. EFNA4 was previously studied in other epithelial cancers, but first time studied in lung cancer.

Given the poor prognosis of metastatic lung cancer, it was a nice study to show a novel gene implicated in the migration of lung cancer cells and specific to metastatic stages. This could constitute a potential target for better effective treatments of metastatic lung cancer.

Strengths:

-       Very concise detailed introduction providing a thorough description of the literature.

-       Study of both overexpression/ knock-out by CRISPR and siRNA in vitro and in vivo.

-       In vitro: they have shown that overexpressing of EFNA4 increases the proliferation and migration of tumor cells. Results validated in vivo; the overexpression increases the tumor volume and tumor weight comparing to the control.

-       In vitro: the KO of EFNA4 decreases the proliferation and the migration, the same results validated in vivo, where they have shown the KO have less tumor volume.

Comments:

-    Figure 1B-C, what is the data source? Should be written in the legend (TCGA project)

-     Why is HEK293 described in Fig2A? The cell line does not serve a purpose in the manuscript since it is not used anywhere else. If it is describing the expression of the vector, then this is shown by the lung carcinoma cell lines.

-     Could there be a better Western Blot figure for A549 and PC9 in Fig. 2A?

-     Mistake in Figure 2C and D labeling

-     Could 2B and C be quantified or at least labelled as representative figures in the legends?

-     Fig 2C could be enlarged

-     What is defined as “Wild Type” cells for Fig, 2E? Are they stably transfected with an empty vector?

-     Figures are missing a description for the asterisks on the bar graphs

-     If EFNA4 could be a target for therapy, as stated by authors, then how would they envision improving the PF-06647263 AB treatment for lung cancer? Could authors describe other potential methods to block EFNA4 expression.

-     Though the authors clearly show EFNA4 affecting activities of metastasis, i.e. proliferation and migration, the authors may want to replace the word metastasis from their manuscript since they do not show the development of secondary malignant growths at a distance from a primary site of cancer.

Author Response

Point 1: Figure 1B-C, what is the data source? Should be written in the legend (TCGA project)

Response 1: Thanks for this point. The source of the data has now been carefully described in the Figure legend.

Point 2: Why is HEK293 described in Fig2A? The cell line does not serve a purpose in the manuscript since it is not used anywhere else. If it is describing the expression of the vector, then this is shown by the lung carcinoma cell lines.

Response 2: Thanks for this point, we have removed HEK293 in Fig2A.

Point 3: Could there be a better Western Blot figure for A549 and PC9 in Fig. 2A?

Response 3: We have updated this figure. Because EFNA4 is overexpressed at a high level, it is difficult to observe the endogenous EFNA4 expression in control cells.

Point 4: Mistake in Figure 2C and D labeling

Response 4: Thank you for the reminder. We have reordered fig 2C and 2D in the revised manuscript.

Point 5: Could 2B and C be quantified or at least labelled as representative figures in the legends?

Response 5: Thank you for the reminder. We have quantified the number of clones and migration and revised the legend. We replace "WT" with "control" and "EFNA4" with "OE".

Point 6: Fig 2C could be enlarged

Response 6: Figure 2c has been enlarged in the revised manuscript.

Point 7: What is defined as “Wild Type” cells for Fig, 2E? Are they stably transfected with an empty vector?

Response 7: As you mentioned, we did use control cells transfected with an empty vector instead of wild type. This error has been corrected in revised manuscript.

Point 8: Figures are missing a description for the asterisks on the bar graphs

Response 8: The description of the asterisk has been added in the revised manuscript.

Point 9: If EFNA4 could be a target for therapy, as stated by authors, then how would they envision improving the PF-06647263 AB treatment for lung cancer? Could authors describe other potential methods to block EFNA4 expression.

Response 9: We speculate that there may be the following directions that could improve the clinical efficacy of PF-06647263. 1, the identification of biomarkers that are predictive of clinical efficacy of PF-06647263. Not only EFNA4, but the levels of proteins downstream of EFNA4 may also be related to the efficacy of PF-06647263. 2. In this study, we found that high expression of EFNA4 was significantly associated with lymph node metastasis in TCGA lung cancer patients. PF-06647263 may be more effective in patients with lymph node metastases. In addition to antibody-drug conjugates (ADCs), targeted protein degradation and CRISPR are also very promising for clinical treatment. 

Point 10: Though the authors clearly show EFNA4 affecting activities of metastasis, i.e. proliferation and migration, the authors may want to replace the word metastasis from their manuscript since they do not show the development of secondary malignant growths at a distance from a primary site of cancer.

Response 10: Thanks for this very important suggestion. In our study, we combined clinical and gene expression data from TCGA and found that EFNA4 is closely related to lymph node metastasis in lung cancer. As the reviewer point out we do not provide experimental evidence to directly support the function of EFNA4 in cancer metastasis. This limitation has now been discussed in the revised discussion part.

Reviewer 2 Report

In the current research article, " Oncogenic EFNA4 amplification promotes lung adenocarcinoma lymph node metastasis" the authors argue about EFN4 as a possible target for metastatic cancer therapy. After performing genome-wide analysis authors found that EFN4 mRNA expression levels were significantly elevated in lung cancer patients with lymph node metastases. It is a good experimental design to perform in vitro overexpression and knock-out experiments to prove the involvement of EFN4 in tumorigenesis. Further, in xenograft models, they showed the EFN4 knock-out mice exhibit reduced tumor growth compared to ingrafts from the parental cell lines. Overall, the research approach is good but, I encourage authors to dwell on to metastasis issue as well. Did the authors look at changes in metastatic potential following the KO or overexpression of EFN4? This is one of the critical experiments that could potentially improve the impact of the manuscript. I think the authors need to perform a few critical in vivo experiments to confirm the importance of EFN4 in metastasis/tumorigenesis.

Comment

Comment 1: Kindly look at metastatic burden in EFN4 KO and overexpression mice. This will prove the involvement of EFN4 in metastasis. If possible, using luciferase expressing lines will be beneficial to track the metastasis in the EFN overexpression and KO models.

Comment 2: Basal Expression levels of EFN4 in the parent (non-transduced/ non- transfected) A549 and PC9 cell lines from Figure-2 and Figure-3 do not match.

Comment 3: Make sure the unit of tumor mass is correct. (0.3mg) tumor with 600mm3 volume is not possible

Comment 4: Can authors confirm EFN4 protein expression from tumor explants used in experiments in Figure 2E and 3F?

Comment 5: Also, it will be interesting to investigate changes mRNA signature in tumors with overexpression/knock-out of EFN4.

Comment 6: It will also be useful if authors can do survival study using EFN4 overexpression or KO in mouse models and see if they observe any correlation with the human patient data shown in figure 4.

Minor Comments:

1.       Please adjust the contrast and resolution for the scratch assay. Can be made enlarged.

Author Response

Point 1: Kindly look at metastatic burden in EFN4 KO and overexpression mice. This will prove the involvement of EFN4 in metastasis. If possible, using luciferase expressing lines will be beneficial to track the metastasis in the EFN overexpression and KO models.

Response 1: Thanks for this important suggestion. As the reviewer pointed out that we do not have experimental evidence to directly support the function of EFNA4 in cancer metastasis, and this limitation has been discussed in the discussion part. Experimental system for studying cancer metastasis in vivo is not well established yet, and perform these additional experiments will be our future works.

Point 2: Basal Expression levels of EFN4 in the parent (non-transduced/ non- transfected) A549 and PC9 cell lines from Figure-2 and Figure-3 do not match.

Response 2: This is because EFNA4 is overexpressed at a high level in H1299 cells and this makes it difficult to observe the endogenous expression of EFNA4 in control cells.

Point 3: Make sure the unit of tumor mass is correct. (0.3mg) tumor with 600mm3 volume is not possible

Response 3: Thanks for this point. The units have now been carefully checked and confirmed.

Point 4: Can authors confirm EFN4 protein expression from tumor explants used in experiments in Figure 2E and 3F?

Response 4: Due to following reasons, 1. The tumor tissue developed by the knockout cells is too small; 2. A portion of the tissue has been used in other experiments; 3. The specificity of the EFNA4 antibody is not very good, so it is difficult for us to extract sufficient protein from the remaining tissues and used for WB verification. Therefore, we can only use the remaining tissues to extract RNA and performed qPCR to compare the mRNA levels of EFNA4 in the overexpressed tissues relative to those in the control group. The results have been added to the figure S3. Real-time qPCR showed that mRNA levels of EFNA4 was near 4.5-fold higher in EFNA4 over-expressing cells form tumor xenografts compared to empty control cells.

Point 5: Also, it will be interesting to investigate changes mRNA signature in tumors with overexpression/knock-out of EFN4.

Response 5: Thanks for this point. It is indeed very meaningful to explore the changes of gene expression in cells after knockout or overexpression of EFNA4. As we mentioned in the "discussion" part, we will perform in-depth exploration in the future to check the potential mechanism involved.

Point 6: It will also be useful if authors can do survival study using EFN4 overexpression or KO in mouse models and see if they observe any correlation with the human patient data shown in figure 4.

Response 6: Thanks for this suggestion. Survival analysis with nude mice bearing tumor probably cannot validate the finding in Figure 4, since the mice would be sacrifices after the size of tumor reach the limit. Tumors derived from EFNA4 knockout mice grow slowly and will consequently have longer survival compared with nude mice bearing control tumors. An appropriate validation experiment would be using ENFA4 knockout or transgenic mouse model and cross these mice with known lung cancer mouse models (such as KRAS G12D mouse model), then evaluate the survival of these mice. However, construction of EFNA4 knockout or transgenic mouse model is beyond the scope of this study.

Point 7: Please adjust the contrast and resolution for the scratch assay. Can be made enlarged.

Response 7: Thanks for this point. We have reprocessed the image to make it clearer and larger.

Round 2

Reviewer 2 Report

The authors answered most of my questions in the current version. However, I believe, that using the word "Metastasis" in the title without backing it up with experimental data is unwarranted.  Authors need to either revise the title or show some experimental data on connecting EFN4A function with metastasis. Authors can look at certain genes that are commonly involved in metastasis using the tumors used in EFN4A overexpression or knockout studies.